# 3D Networks of Ge Quantum Wires in Amorphous Alumina Matrix

**DOI:** 10.3390/nano10071363

**Published:** 2020-07-13

**Authors:** Lovro Basioli, Marija Tkalčević, Iva Bogdanović-Radović, Goran Dražić, Peter Nadazdy, Peter Siffalovic, Krešimir Salamon, Maja Mičetić

**Affiliations:** 1Rudjer Boskovic Institute, 10000 Zagreb, Croatia; lovro.basioli@irb.hr (L.B.); marija.tkalcevic@irb.hr (M.T.); iva@irb.hr (I.B.-R.); Kresimir.Salamon@irb.hr (K.S.); 2National Institute of Chemistry, 1001 Ljubljana, Slovenia; goran.drazic@ki.si; 3Institute of Physics, Slovak Academy of Sciences, 845 11 Bratislava, Slovakia; peter.nadazdy@savba.sk (P.N.); peter.siffalovic@savba.sk (P.S.)

**Keywords:** Ge quantum wires, 3D ordering, self-assembly, quantum wire network, quantum confinement

## Abstract

Recently demonstrated 3D networks of Ge quantum wires in an alumina matrix, produced by a simple magnetron sputtering deposition enables the realization of nanodevices with tailored conductivity and opto-electrical properties. Their growth and ordering mechanisms as well as possibilities in the design of their structure have not been explored yet. Here, we investigate a broad range of deposition conditions leading to the formation of such quantum wire networks. The resulting structures show an extraordinary tenability of the networks’ geometrical properties. These properties are easily controllable by deposition temperature and Ge concentration. The network’s geometry is shown to retain the same basic structure, adjusting its parameters according to Ge concentration in the material. In addition, the networks’ growth and ordering mechanisms are explained. Furthermore, optical measurements demonstrate that the presented networks show strong confinement effects controllable by their geometrical parameters. Interestingly, energy shift is the largest for the longest quantum wires, and quantum wire length is the main parameter for control of confinement. Presented results demonstrate a method to produce unique materials with designable properties by a simple self-assembled growth method and reveal a self-assembling growth mechanism of novel 3D ordered Ge nanostructures with highly designable optical properties.

## 1. Introduction

Semiconductor nanowires represent one of the most powerful and adaptable classes of building blocks for new functional materials and devices. They have a specific geometry that, due to quantum confinement, strongly influences their opto-electronic properties including quantum transport, which is important for modern nanotechnology devices [1,2,3,4,5]. Therefore, nanowires are promising for applications in fields such as electron devices, quantum computing, optoelectronics, sensing devices, and many others [6,7,8].

Particularly interesting are networks of quantum wires as they act like artificial solids because their properties are determined by the structure of nano-scale building blocks and their arrangement [9,10,11]. They are of great technological importance for various applications including electro-catalysis, sensitive sensing, and improvement of electronic devices [9,10].

Germanium quantum wires (QWs) have attracted a lot of attention due to their high mobility of electrons and holes, promising faster switching and application in higher frequency devices, higher intrinsic carrier concentrations, good compatibility with high-dielectric-constant materials, and a large exciton radius that enables strong confinement effects in relatively large structures [12,13]. Ge QWs are mostly grown by chemical vapor deposition, laser ablation, supercritical fluid–liquid–solid synthesis, thermal evaporation, or template method [14]. However, QWs produced by these methods are usually not arranged in a desirable 3D-regular structure. The realization of 3D-ordered QW structures often requires expensive and time-consuming processes like high-resolution lithography for defining patterns at the nanometer scale, and chemically or electrically-driven assembly [9]. Aside from the complicated production procedure, such structures often suffer from weak QW connectivity at network nodes. Consequently, production methods for obtaining 3D-ordered QW structures by self-assembly processes are of great importance.

Here, we present a material consisting of germanium QWs ordered in a 3D network within an alumina matrix, produced by self-assembling growth during the magnetron sputtering co-deposition of Ge and alumina (Al_2_O_3_). We explore the dependence of the geometrical properties of these networks on deposition conditions. The results show the tenability of the structural parameters of these 3D networks, which are easily controllable by the deposition conditions including Ge concentration and deposition temperature. The geometrical properties of QW networks as well as QW radii follow simple rules, enabling their controllable production. We show an interesting property of the network geometry to adopt an increase in Ge concentration during deposition by tilting the angle of QWs toward the surface. The QWs increase the tilt angle with Ge concentration, keeping their radius nearly constant, and adjust the network geometrical parameters to receive the excess of Ge. On the other hand, the deposition temperature controls the QW radii and their separation. These simple mechanisms enable the manipulation of the geometrical properties of the QW networks in a really broad range, and consequently their opto-electrical properties. The prepared materials, besides excellent interconnectivity, exhibit strong confinement effects, clearly visible in their optical properties. Interestingly, the confinement effects showed a trend that seems to be the opposite of that expected; the energy shift was larger for longer QWs with similar radii. In fact, the strongest confinement was observed for the longest QWs due to the existence of QW network nodes, in which four QWs joined together, increasing the actual radius significantly. Separation between these nodes was the largest for the longest QWs, enabling observation of the strongest confinement.

The presented material can be used for application in modern nanotechnology devices due to the designable charge transport explored in our previous work [11] and tunable optical transmission explored here.

## 2. Materials and Methods 

Growth of QW networks was achieved by the co-deposition of Ge and Al_2_O_3_ using the magnetron sputtering KJLC CMS-18 system, produced by Kurt J Lesker Company Ltd. We have used Ge (99.999%) and Al_2_O_3_ (99.999%) targets (Kurt J Lesker Company Ltd.) The thin films were deposited on quartz and Si(100) substrates (produced by University Wafer Inc., Boston, USA) at temperatures in a range from room temperature (RT) to 600 °C. Ge sputtering power was tuned in a range from 2.5 to 30 W, while the power of Al_2_O_3_ sputtering was kept constant at 140 W, except for one case in which the power was 200 W to ensure low Ge concentration in the film. The minimal and maximal powers for each target were determined by the producer. Argon pressure was 3 mTorr for all films. The substrates were rotated at 1 rpm during deposition to ensure homogeneous deposition of the films. Main deposition parameters and film names are given in Table 1. The film names are comprised of two letters indicating sputtering power (P) and deposition temperature (T), followed by a number related to the index of the pressure or temperature. The described deposition procedure resulted in films consisting of a 3D network of Ge QWs embedded in an Al_2_O_3_ matrix. Concentration of Ge in the films (in at. %), given in the last row of Table 1, was measured by the Time-of-Flight Elastic Recoil Detection Analysis (TOF ERDA) technique (Ruđer Bošković Institute, Zagreb, Croatia). It was found that concentrations of Ge were nearly the same for a constant Ge sputtering power. The largest deviations were found for the highest deposition temperature, for which the films were not fully homogeneous.

Grazing Incidence Small Angle X-ray Scattering (GISAXS) patterns were measured using a custom-designed x-ray scattering setup (Bruker AXS GmbH, Karlsruhe, Germany). The setup was equipped with a liquid–metal jet anode x-ray source MetalJet D2+ (Excillum AB, Kista, Sweden) emitting at the wavelength of 1.34 Å. Beam collimation was performed by a parallel Montel optics (Incoatec GmbH, Geesthacht, Germany) and two 550 µm scatterless Ge pinholes (Incoatec) 50 cm apart. The scattered x-rays were collected by a two-dimensional hybrid pixel detector Pilatus 300 K (Dectris AG, Baden, Switzerland). The samples were aligned using a positioning hexapod (Physic Instrumente) platform placed in an evacuated chamber. 

The GISAXS data have been analyzed using home-made program written in Matlab.

Germanium concentration (in at.%) was determined by TOF ERDA measurements, performed using the TOF ERDA spectrometer [15,16] attached to the 0° beam line at the Ruđer Bošković Institute accelerator facility. The angle between the incoming 20 MeV ^127^I^6+^ ions and the sample surface was 20°. The scattered and recoiled ions were detected under 37.5° with respect to the incident ion beam. The uncertainty in the reported elemental concentrations was estimated to be around 8%.

Transmission measurements were carried out using Ocean Optics equipment including a deuterium-halogen light source (DH-2000-BAL), a UV/VIS detector (HR4000), and SpectraSuite software.

Scanning transmission electron microscopy (STEM) was performed using a probe Cs corrected JEOL ARM 2000 CF scanning transmission electron microscope, operated at 200 kV, and equipped with a high-angle annular dark-field detector (HAADF) for Z-contrast imaging.

## 3. Results

The structure of 3D QW networks is described in this section including the arrangement of their nodes, QW radii, length, and their dependence on deposition conditions. 

### 3.1. Structural Properties

#### 3.1.1. Quantum Wire Network Structure

A typical structure of the prepared materials, imaged by STEM and GISAXS, is demonstrated in Figure 1. The microscopy images of the materials prepared under different conditions (different Ge concentrations in Figure 1a,b, and different temperature and Ge concentration in Figure 1c) all showed ordering in a 3D network, as schematically shown in Figure 1d. The ordering type was a body-centered tetragonal (BCT) lattice, as shown in [11]. The same type of ordering was found for Ge quantum dots in an alumina matrix, grown by a very similar process [17]. Depending on the deposition conditions, the QWs make networks with different geometrical parameters. Smaller Ge sputtering powers (smaller Ge concentrations) lead to a larger unit cell of the network, which follows from the comparison of networks in Figure 1a,b. 

The same structure was seen from the GISAXS maps of the films, as shown in the lower right corner of the main STEM images, in which Bragg spots (lateral intensity peaks) are well resolved. The shape and position of the spots are closely related to geometrical parameters of QW networks [10,11]. The GISAXS technique shows the structure in reciprocal space, therefore, a smaller separation between Bragg spots encodes a larger separation of the corresponding QW network nodes, and narrower Bragg spots reflect a better quality of 3D ordering. Thus, network T5P2 (Figure 1a) had the largest network parameters and the best quality of ordering, as visible from its microscopy image, and the most separated as well as the narrowest Bragg spots in the corresponding GISAXS map. On the other hand, the smallest network parameters and the highest disorder with respect to the ideal lattice were found for film T2P7 (Figure 1c), which also had the most separated and very elongated (almost circular-like) Bragg spots. A detailed description of properties of GISAXS maps from nanowire networks can be found in [18,19].

To explore the structural properties of QW networks, we deposited films using broad ranges of Ge sputtering power and deposition temperatures. The resulting Ge concentrations varied approximately from 5 to 70 atomic percent in the material, while the deposition temperature varied from room temperature to 600 °C, which is the limiting temperature for the deposition system. For very high Ge concentrations, the QWs were very close to each other and overlapped significantly, so we actually had alumina nanoparticles in Ge. On the other hand, for a very low Ge concentration, the interconnectivity of the QWs was lost. The GISAXS maps of all prepared films are shown in Figure 2. From Figure 2, it is visible that an increase in Ge concentration led to a broadening of the characteristic semi-circular signal consisting of elongated Bragg spots, showing a decrease in the unit cell of the QW network. In addition, a weak increase in the width of the spots showed an increase of the disorder in the QW networks. On the other hand, an increase in the deposition temperature led to a sharpening of the Bragg spots and narrowing of their separation, indicating an improvement in the quality of the QW ordering in a 3D network, followed by an increase in the QW network parameters. Films T5P1, T6P1, and T6P2 (both high deposition temperature and low Ge concentration) did not show a GISAXS signal, meaning that they did not consist of ordered Ge QWs. As above-mentioned, the films deposited at 600 °C were not homogeneous for the lowest Ge concentration, although the substrate was rotated during the deposition. This confirms that this is the limiting temperature for the production of QW networks with low Ge content. This can be due to the fact that Ge atoms do not make the QW networks at this temperature, or that they do not adhere well to the substrate during the deposition. 

All GISAXS maps were fitted using the procedure described in [18,19]. The body centered tetragonal (BCT )structure of the network nodes was assumed, and fitting parameters were the QW network unit cell parameters, radius of the QWs, and their statistical distributions. The unit cell structure and parameters are shown in Figure 3, together with the main results of the fit. The parameters obtained from the GISAXS analysis were fitted using the 2D second-order polynomial fit to obtain smooth dependence of the parameters on the deposition conditions. The fit results are also shown in Table 2. The data from the samples without Ge ordering were excluded from the fitting process. From Figure 3a, it follows that the in-plane (parallel to the substrate) unit cell parameter *a* increased with the deposition temperature, and it decreased with the *c*_Ge_ increase. Therefore, the highest *a* parameters were obtained for the lowest concentrations and the highest temperatures. A very similar trend was observed for the vertical (perpendicular to the substrate) separation between the nodes *c* (Figure 3b). Interestingly, the QW diameter did not change significantly with Ge concentration change, while it increased with the deposition temperature (Figure 3c). This finding is in accordance with our previous study where the dependence of Ge quantum dot sizes on the deposition temperature was investigated [17]. Finally, we calculated the length of the QWs from the parameters *a* and *c*, which is shown in Figure 3d. The extrapolated data for the three samples without nanostructure (maximum *a* and *c* values) suggest that there is a geometrical limit for network formation, which could originate from the lack of correlation between QW nod positions due to their distance.

Simulations of the selected QW networks, assuming no disorder, using the parameters of GISAXS fits, are shown in Figure 4. A gradual change of the structure was observed as a decrease in the unit cell with the Ge concentration and its increase with the deposition temperature. As visible from Figure 4, a broad range of network parameters could be obtained using the above-described procedure.

#### 3.1.2. Quantum Wire Network Growth

Here, we try to explain the reasons leading to the observed growth of QW networks. The BCT-structure of the networks was the consequence of the surface morphology effects as explained and simulated in [17,20,21]. However, the gradual change of QW network parameters and its dependence on Ge concentration and deposition temperature needs to be clarified. Therefore, we analyzed in detail the geometrical properties of QW networks. Figure 5 summarizes the most important findings. Figure 5a demonstrates the parameters of the network important for understanding its growth properties including tilt-angle α between the QW direction and the plane parallel to the substrate (shaded area in Figure 5). Dependence of the tilt angle α on the growth conditions (Figure 5b) shows that it depends strongly on the Ge concentration, and only weakly on the deposition temperature. It seems that QWs grow in a way to adjust their tilt to collect all deposited Ge, without changing the ordering type. To better understand this effect, properties of the growing surface (layer parallel to the substrate) should be considered first. The interesting property is the effective radius *R*_eff_ of the QWs in the direction along their tilt, in the plane parallel to the substrate of the film (Figure 5c). Due to the tilt of the QWs, the *R*_eff_ is larger than *R*, and it increases with the decrease of tilt angle α. Dependence of this parameter on the deposition conditions is also shown in Figure 5c. This parameter is important for the calculation of the ratio of the surface covered by Ge with respect to the Al_2_O_3_ matrix. This ratio should follow the Ge concentration increase. The calculation of the ratio is shown in Figure 5d together with a scheme for its calculation. From the figure, it is visible that this ratio is constant for a particular Ge concentration, and it increases with the increase in concentration. Only the samples with the highest amount of Ge were exceptions, probably because the Ge concentration was too high.

This means that Ge QWs indeed adjust their tilt angle to accommodate Ge concentration increase. We calculated the Ge concentration from the geometrical parameters of the networks found by the GISAXS analysis, the results of which are shown in Figure 6, together with the measured Ge concentration. From the figure, it is clear that the concentrations were nearly the same, except for the highest Ge concentration, which was already discussed to be too large, so the model used for their description was no longer a good approximation.

In summary, the QWs adjust their tilt-angle to accommodate all Ge atoms reaching the surface during the deposition. The basic type of the QW ordering does not change, but the network parameters change in a broad range, controllable by deposition temperature and Ge concentration.

### 3.2. Optical Properties

In this section, we explore the main optical properties of the prepared QW networks. The most important properties are shown in Figure 7. The transmissivity of the films deposited at 400 °C (T4) for all prepared Ge concentrations (P1–P8) is shown in Figure 7a. The overall transmissivity decreased gradually with the increase in Ge concentration, as expected due to the increase in Ge concentration; Ge absorbs strongly in the measured range, while the alumina matrix is practically transparent. As we were interested in the quantum confinement effects, we concentrated on the section of the curves where transmissivity went to zero. To explore the confinement effects, we normalized the transmission curves to the same Ge concentration, because the alumina matrix had practically no absorption in this range, so it could be neglected. The normalized graphs plotted as a function of photon energy are shown in Figure 7b. A significant shift of the energy for which the transmissivity went to zero was clearly visible. This energy shifted toward larger values with a decrease in Ge concentration, indicating the existence of strong confinement effects in these films. The essential parameters for the confinement effects, QW radius, and length, are shown in Figure 7c. The QW radii were nearly constant, while the QW length significantly decreased. This strongly suggests that the confinement effects were governed by the QW length in this case.

However, smaller Ge nanostructures should cause stronger quantum confinement and accordingly, larger band gap of the material. However, our results indicated the opposite trend, that is, longer QWs had a larger cut-off energy for absorption, instead of shorter ones. Additionally, the QWs were interconnected, so their length was in fact significantly longer than the Ge exciton radius and should not affect the confinement. Since the QW radii were similar in all samples, one would expect similar optical properties. Again, our results clearly showed that this was not the case, so a hypothesis was made where only a fraction of the Ge in the sample was confined in two directions, while the others were different. More precisely, the nods of the QW network had extensions in eight directions (please see Figure 5a), and four QWs joined together from each side of the node. Therefore, their radii changed significantly in these points, and in fact, we had a complex shape nanoobject instead of an isolated wire or dot. We do not go into detailed calculations of the confinement effects for these structures in this paper, but can make simple conclusions from the experimental data. Thus, in the vicinity of the nod, the confinement effects are altered due to the joining of eight QDs, and consequently, they are weaker due to the increase in radius. Although this nano-geometry can still affect the electronic structure, the effect is greatly overshadowed by the effect of confinement in Ge outside the node. Therefore, we considered the Ge outside the nods as quantum confined. The ratio of confined and unconfined Ge is connected to the QW length *L*, and therefore the optical properties strongly depend on it. The larger the *L* (in samples with the low amount of Ge), the larger the share of confined Ge and, therefore, weaker absorption in the bandwidth between the bulk Ge band gap (around 0.7 eV) and confined Ge band gap. In our case, the largest band gap was about 3.6 eV for the longer QWs with a diameter of about 1 nm. This is supported by the observed regions of the spectrum with a rapid decrease in absorption. More precisely, near the band gap, the absorption increased rapidly due to a rapid increase of available states and we could observe two such regions. One was near the bulk Ge band gap and was a feature for the samples with a smaller wire length because of the small share of confined Ge. The other region was near the confined Ge band gap and was featured in the samples with larger wire length and larger share of confined Ge. Overall, we believe that the measured absorption was a superposition of the absorptions of confined and unconfined Ge, with varying ratios of the two. The observed band gap was larger than that predicted for the crystalline Ge QW of this diameter [22,23]. However, it was in full agreement with the confinement in amorphous Ge quantum wells [24], in which a very strong confinement, stronger than that in crystalline Ge quantum systems, was observed.

The same follows from the dependence of the transmissivity on the deposition temperature (the constant Ge concentration P5), as shown in Figure 7d,e. In this case, the concentrations were nearly the same, but the QW length and also diameter were larger for the higher deposition temperature. The energy shift dependence on wire size, again, had the opposite direction than usual, in accordance with the above-given arguments. In addition, we noticed that the shift was significantly smaller when compared to the Ge concentration dependence shown in Figure 7b. This is partly because of the compensation made by the change in the wire diameter, which modified the band gap in the expected way.

## 4. Conclusions

We demonstrated a simple method to grow films with self-assembled Ge quantum wire 3D network in an alumina matrix via magnetron sputtering. A large number of different samples were grown, most of them successfully forming the networks with the same type of structure with their nods located in a BCT lattice. In addition, we explored the influence of deposition conditions such as temperature and Ge concentration on the structural parameters of the forming networks. The radii of the wires and their in-plane separations were mainly defined by the deposition temperature, similar to those of the already studied Ge quantum dots. However, the vertical separation was mainly defined by Ge concentration, which was a consequence of the adjustment of the network tilt angle to adopt all available Ge while maintaining the same structure type. This effect implies numerous possibilities for the application of these materials. The optical measurements shown above demonstrate the quantum confinement effect, which can be exploited to design sensing or photovoltaic devices. The materials should also be considered in accordance with their transport properties as investigated in our previous work, which demonstrated different conduction mechanisms in similar films. Taking everything into account, one can consider implementing these materials for a wide variety of applications.

## Figures and Tables

**Figure 1 nanomaterials-10-01363-f001:**
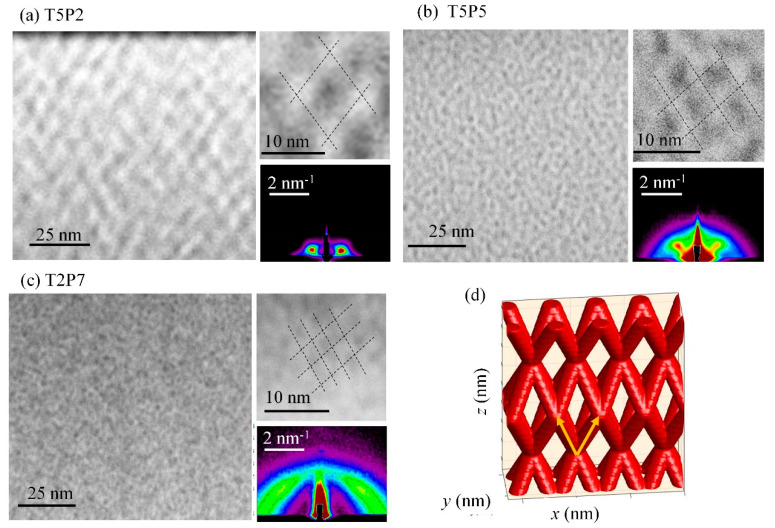
Typical structure of three-dimensional (3D) Ge nanowire networks. (**a**–**c**) a high-angle annular dark-field detector - Scanning transmission electron microscopy (HAADF-STEM) images of the films’ cross-sections. The insets show an enlarged part of the microscopy images, and the grazing incidence small angle x-ray scattering (GISAXS) map of the corresponding films, (**d**) schematically present the ideal structure of quantum wire (QW) networks with no disorder.

**Figure 2 nanomaterials-10-01363-f002:**
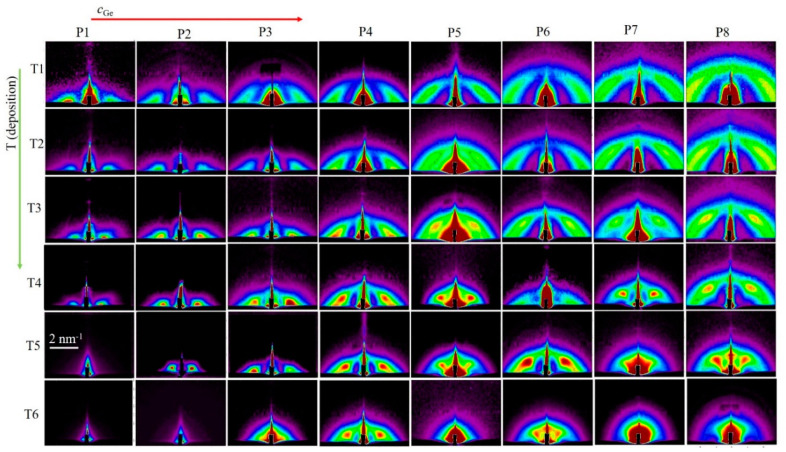
GISAXS maps of all investigated films. An increase in Ge sputtering power (index P), which determines Ge concentration in the film, and deposition temperatures (index T) are indicated at the top and at the left side of the figure, respectively. Numerical values for parameters are given in Table 1.

**Figure 3 nanomaterials-10-01363-f003:**
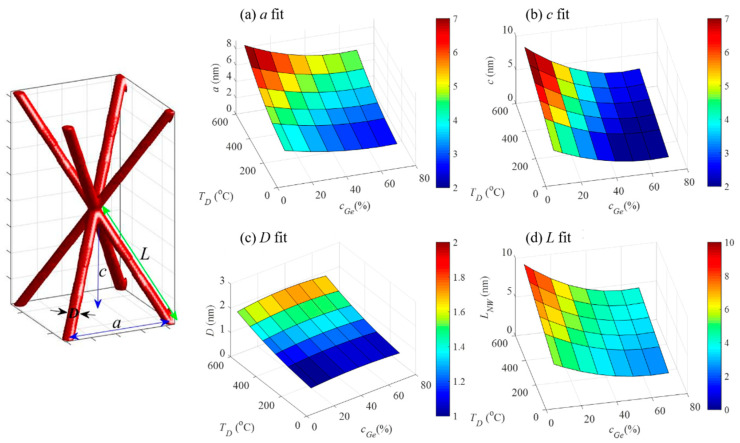
Dependence of the structural parameters of QW networks including (**a**) in-plane separation *a*, (**b**) vertical separation *c*, (**c**) QW diameter *D*, and (**d**) QW length *L* on the deposition parameters. The parameters are indicated in the simulated unit-cell structure of the prepared 3D networks with a BCT arrangement of the network nodes, shown in the left.

**Figure 4 nanomaterials-10-01363-f004:**
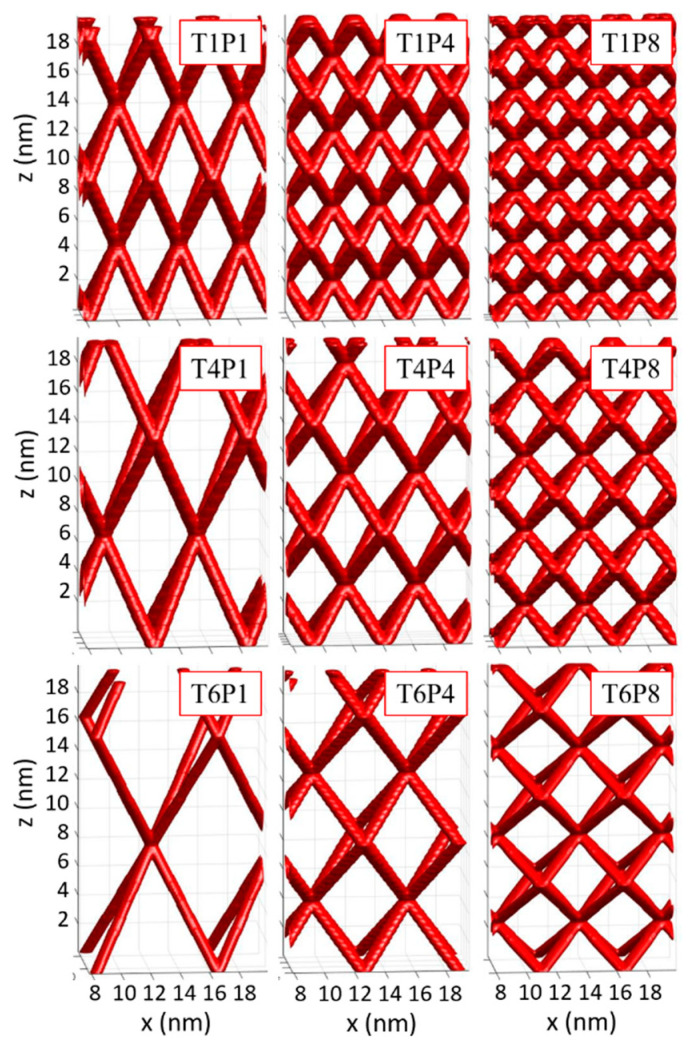
Simulated selected networks of the nanowires obtained from different deposition conditions. The simulations were performed using the parameters obtained from the GISAXS analysis. It was assumed that the structure was ideal with no disorder.

**Figure 5 nanomaterials-10-01363-f005:**
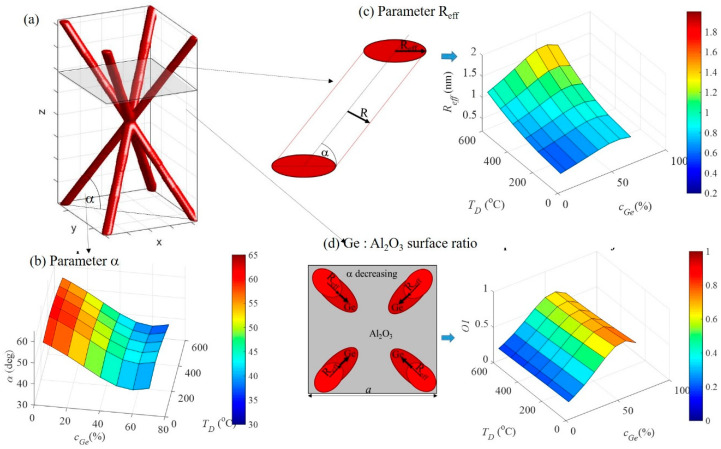
(**a**) Scheme of the QW network unit cell and parameters important for understanding its geometrical properties including tilt-angle α and parallel to surface cross-section of the QWs. Dependence of (**b**) tilt angle α, (**c**) QW effective radius (*R*_eff_), and (**d**) Ge:alumina surface ratio, on the deposition parameters.

**Figure 6 nanomaterials-10-01363-f006:**
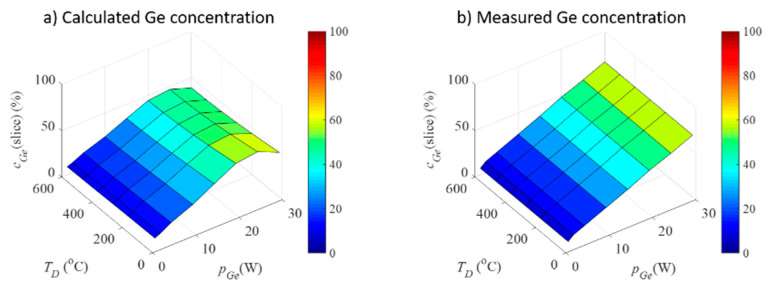
(**a**) Measured and (**b**) calculated from the GISAXS-fit, Ge concentration in dependence on the deposition parameters.

**Figure 7 nanomaterials-10-01363-f007:**
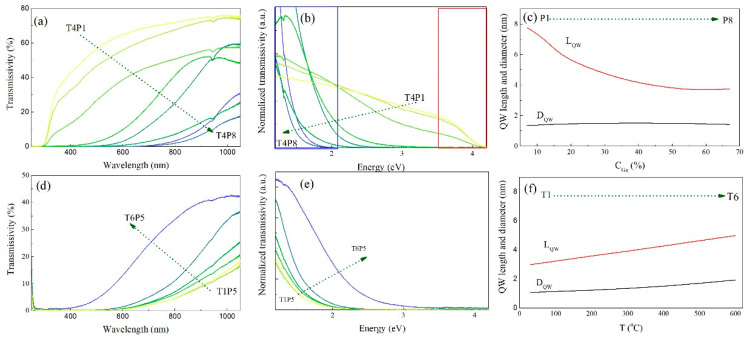
(**a**) Transmissivity, (**b**) normalized transmissivity, and (**c**) geometrical parameters of the materials prepared at 400 °C (T4), but with the varying Ge concentrations P1–P8. The marked areas in (**b**) at smaller and larger energies mark the rapid transmissivity decrease near the band gap energies of the bulk and confined Ge, respectively. (**d**) Transmissivity, (**e**) normalized transmissivity, and (**f**) geometrical parameters of the materials prepared with the same Ge sputter power P5 (15 W), but with the varying deposition temperatures of T1–T6.

**Table 1 nanomaterials-10-01363-t001:** Deposition parameters of the films and Ge concentrations. Sputtering power of Al_2_O_3_ was 140 W for all films except those indicated by * for which it was 200 W. The last row shows the measured concentrations of Ge (at. %) obtained for each sputtering power.

Title	P1 (2.5 W) *	P2 (2.5 W)	P3 (5 W)	P4 (10 W)	P5 (15 W)	P6 (20 W)	P7 (25 W)	P8 (30 W)
T1 (RT)	T1P1	T1P2	T1P3	T1P4	T1P5	T1P6	T1P7	T1P8
T2 (200 °C)	T2P1	T2P2	T2P3	T2P4	T2P5	T2P6	T2P7	T2P8
T3 (300 °C)	T3P1	T3P2	T3P3	T3P4	T3P5	T3P6	T3P7	T3P8
T4 (400 °C)	T4P1	T4P2	T4P3	T4P4	T4P5	T4P6	T4P7	T4P8
T5 (500 °C)	T5P1	T5P2	T5P3	T5P4	T5P5	T5P6	T5P7	T5P8
T6 (600 °C)	T6P1	T6P2	T6P3	T6P4	T6P5	T6P6	T6P7	T6P8
C_Ge_ (%)	7	12	17	27	37	48	57	67

**Table 2 nanomaterials-10-01363-t002:** Fitting results of the dependence of structural parameters on deposition conditions. The shape of every function was the same: $F\left(T_{D},c_{Ge},p_{1},p_{2},p_{3},p_{4},p_{5},p_{6}\right)=\left(p_{1}T_{D}^{2}+p_{2}T_{D}+p_{3}\right)∙\left(p_{4}c_{Ge}^{2}+p_{5}c_{Ge}+p_{6}\right)$; parameters for each function are given in its table row. Data for *a*, *c*, and *D* are taken from the GISAXS map fit.

F	*p*_1_ [10^−6^ K^−2^]	*p*_2_ [10^−5^ K^−1^]	*p*_3_ [10^−4^]	*p*_4_ [10^−2^ nm]	*p*_5_ [nm]	*p*_6_ [nm]
a	2.45	38.8	4.10	−5.04	4.03	1.07
c	400	320	4.05	−4.86	2.07	2.53
D	0.178	5.26	−3.13	2.64	3.96	0.118

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
