# Peer review of "3D Networks of Ge Quantum Wires in Amorphous Alumina Matrix"

_nanomaterials, 2020, doi:10.3390/nano10071363_

Round 1
Reviewer 1 Report
In principle this is a very interesting paper. Results are fancy, analyses are fine, graphics, tables are good. Overall acceptable for publication.
I have issues with the "packaging", the language, though.
For one, the title "... in dielectric matrices" suggests more than it can hold. Effectively, only Al2O3 is explored as matrix –– thus, just one dielectric matrix.
=> this should be changed.
Secondly, there is a clear change in language style. While the body of the text is acceptable, the abstract and introduction is (almost) unbearably over-burdened with buzz words and attention-seeking constructs. That devalues the appeal considerably and may even lead to less attention. I urge the authors to moderate this part –– the paper will be better afterwards.
Once that's done, the manuscript should get published.
A few more details:
The introduction seeks quite some attention by adding a couple of "high profile terms" (aka "buzz words"). In my opinion, this attention-seeking is overdoing it, it even creates nonsense terms. Less will be more.
For example:
p1 l31: "adaptable classes of the artificially designable building blocks"
* explain a difference between "designable" and "artificially designable" ?
Why not simple: "adaptable classes of building blocks" ? (and without article "the")
p1 l32: "They have a specific, confined geometry that"
* strictly speaking, a wire has no "confined geometry". It is a 1D-object. Electrons in a nanowire may experience "confinement" ...
There is also a frequent battle with articles (understandably), especially with "the" – where either an undetermined article or none should be used. See the example above, or:
p1 l34: "the nanowires are promising" –– just nanowires
p1 l37: "Especially interesting are the networks of quantum wires" –– just networks
Continuing that sentence:
"as they represent an artificial solid which properties are determined by the properties of the nano-scale building blocks and their geometry"
(cut the "the" again"
* hopping from plural (networks) to singular "an artificial solid"
However, what is an "artificial solid" ? Man made, not occurring in nature ? => then this is nothing special –– there are many compounds. Moreover, whether those networks by themselves are already the "solids" the authors think about, is in question – because the embedding into a matrix is of considerable importance (and part of the study).
More of attention seeking
p2, l56: "The results show incredible tenability"
* this is not a fun fair, but a scientific communication. "incredible" certainly doesn't belong here.
There are more of these constructs ... it is just too tedious to list them all.
Why are "TOF ERDA measurements" mentioned twice in the "Materials and Methods" section?
Author Response
We thank the Referee for the useful comments that helped to improve our manuscript.
- For one, the title "... in dielectric matrices" suggests more than it can hold. Effectively, only Al2O3 is explored as matrix –– thus, just one dielectric matrix.
- The title has been changed to „3D networks of Ge quantum wires in amorphous alumina matrix“
- Secondly, there is a clear change in language style. While the body of the text is acceptable, the abstract and introduction is (almost) unbearably over-burdened with buzz words and attention-seeking constructs. That devalues the appeal considerably and may even lead to less attention. I urge the authors to moderate this part –– the paper will be better afterwards.
- changes: - abstract: - very strong -> strong
- great possibilities -> a method
- show incredible -> reveal
- introduction - is an excellent candidate -> can be used
- strongly tunable -> tunable
- incredible tenability -> tenability
- The introduction seeks quite some attention by adding a couple of "high profile terms" (aka "buzz words"). In my opinion, this attention-seeking is overdoing it, it even creates nonsense terms. Less will be more.
p1 l31: "adaptable classes of the artificially designable building blocks"
* explain a difference between "designable" and "artificially designable" ?
Why not simple: "adaptable classes of building blocks" ? (and without article "the")
- suggestion accepted
p1 l32: "They have a specific, confined geometry that"
* strictly speaking, a wire has no "confined geometry". It is a 1D-object. Electrons in a nanowire may experience "confinement" ...
- suggestion accepted; confined geometry that... -> geometry that, due to quantum confinement,...
There is also a frequent battle with articles (understandably), especially with "the" – where either an undetermined article or none should be used. See the example above, or:
p1 l34: "the nanowires are promising" –– just nanowires
p1 l37: "Especially interesting are the networks of quantum wires" –– just networks
Continuing that sentence:
"as they represent an artificial solid which properties are determined by the properties of the nano-scale building blocks and their geometry"
(cut the "the" again"
- Changes concerning the article „the“ have been made throughout the article.
* hopping from plural (networks) to singular "an artificial solid"
However, what is an "artificial solid" ? Man made, not occurring in nature ? => then this is nothing special –– there are many compounds. Moreover, whether those networks by themselves are already the "solids" the authors think about, is in question – because the embedding into a matrix is of considerable importance (and part of the study).
- By artificial solid we mean that the material's properties are defined by „artificial atoms“ which is a common term used for quantum dots but can also be used for our „quantum nods“. New sentence is: Especially interesting are networks of quantum wires, as they act like artificial solids because their properties are determined by structure of nano-scale building blocks and their arrangement.
- More of attention seeking
p2, l56: "The results show incredible tenability"
* this is not a fun fair, but a scientific communication. "incredible" certainly doesn't belong here.
- suggestion accepted
- There are more of these constructs ... it is just too tedious to list them all.
- Many style changes have been made in the article and we hope that the new style is acceptable.
- Why are "TOF ERDA measurements" mentioned twice in the "Materials and Methods" section?
- Once they are mentioned just to explain how we filled the last row of Table 1. And the other mention is, of course, the explanation of the method.
Reviewer 2 Report
The manuscript "3D networks of Ge quantum wires in dielectric matrices" gave sufficient details for each section reference with proper literature citation to satisfy the curiosity and address the interests of the readers while highlighting some key findings. The conclusions were quite good with some useful recommendations. In my opinion it is a well-rounded manuscript that can be published in Nanomaterials. Only following minor issues need to be revised prior publication:
- The images in Fog. 1 is not of good contrast/resolution. If possible authors should use good quality images. Also, there is huge open space in this figure. Authors may rearrange to utilize free spaces.
- The axes and markings etc are not properly visible in Fig. 4.
Author Response
- The images in Fog. 1 is not of good contrast/resolution. If possible authors should use good quality images. Also, there is huge open space in this figure. Authors may rearrange to utilize free spaces.
- We have prepared a new figure with better resolution. The figure has also been rearranged to utilize most of free spaces.
- The axes and markings etc are not properly visible in Fig. 4.
- The image has been refined.
Reviewer 3 Report
As continuation of previous works, the authors report a comprehensive study on the deposition conditions leading to the formation of 3D networks of Ge quantum wires in alumina matrix. The subject is of great interest from both fundamental and applied viewpoint and is certainly adequate for Nanomaterials. The employed methods are fine and the discussion section is well addressed. The manuscript reports a substantial body of results and it seems a job well done.